# The Role of Isolated Nasal Surgery in Obstructive Sleep Apnea Therapy—A Systematic Review

**DOI:** 10.3390/brainsci12111446

**Published:** 2022-10-26

**Authors:** Emily Schoustra, Peter van Maanen, Chantal den Haan, Madeline J. L. Ravesloot, Nico de Vries

**Affiliations:** 1Department of Otorhinolaryngology—Head and Neck Surgery, OLVG, 1061 AE Amsterdam, The Netherlands; 2Department of Research and Epidemiology, Medical Library, OLVG, 1061 AE Amsterdam, The Netherlands; 3Department of Oral Kinesiology, Academisch Centrum Tandheelkunde Amsterdam, MOVE Research Institute Amsterdam, University of Amsterdam and VU Amsterdam, 1081 LA Amsterdam, The Netherlands; 4Faculty of Medicine and Health Sciences, Department of Otorhinolaryngology, Head and Neck Surgery Antwerp University Hospital, 2650 Antwerp, Belgium

**Keywords:** obstructive sleep apnea, nasal surgery, nose

## Abstract

**Purpose:** Nasal obstruction is believed to play a significant role in the pathophysiology and management of obstructive sleep apnea (OSA). However, controversy remains about the ability of isolated nasal surgery to improve OSA. The objective of this systematic review is to give an updated overview of the literature on whether isolated nasal surgery can improve OSA subjectively (Epworth Sleepiness Scale (ESS)) and/or objectively (polysomnography (PSG)). **Methods:** A systematic review was performed searching the electronic databases PubMed, Embase.com (accessed on 20 June 2022) Cochrane Database of Systematic Reviews and Cochrane Central Register of Controlled Trials (CENTRAL) up to 20 June 2022. Eligible studies were reviewed for methodological quality using the NIH Quality Assessment Tool for Observational Cohort and Cross-Sectional Studies. **Results:** Twenty-one studies met the inclusion criteria. The majority of the included studies reported no significant reduction in the apnea–hypopnea index (AHI) after isolated nasal surgery in patients with OSA. The meta-analysis suggests that the AHI slightly decreases after nasal surgery. The ESS was significantly lower after nasal surgery in eighteen studies. **Conclusion:** Based on the present analysis of objective outcomes, isolated nasal surgery did not improve the AHI significantly in the majority of the studies. The meta-analysis suggests a slight decrease in AHI after nasal surgery, but this reduction is not clinically relevant in terms of treatment success. Isolated nasal surgery should therefore not be recommended as a first-line treatment for OSA. Because of high study heterogeneity, these results should be interpreted with caution. Isolated nasal surgery can possibly improve OSA subjectively. Perhaps only OSA patients with complaints of nasal obstruction or OSA patients experiencing difficulty with continuous positive airway pressure (CPAP) compliance would benefit from isolated nasal surgery.

## 1. Introduction 

Nasal obstruction is common in patients with obstructive sleep apnea (OSA) and is a known risk factor [1]. Nasal obstruction can be caused by nasal congestion due to chronic rhinosinusitis with or without nasal polyps, allergies or structural abnormalities, such as a deviated nasal septum and enlarged turbinates or adenoid. Nasal obstruction caused by nasal packing can also induce apnea in healthy people and also worsen OSA [2,3]. It is therefore recommended that obstructive nasal packing not be used after nasal surgery in patients with OSA [4]. OSA is primarily caused by recurrent episodes of partial or complete obstruction of the upper airway. Pathophysiologically, nasal obstruction causes increased airway resistance, which contributes to oropharyngeal collapse as seen in OSA patients, according to the Starling resistor model [5]. Additionally, mouth breathing as a result of nasal obstruction leads to increased airway resistance [6]. Consequently, nasal obstruction is believed to play a significant role in the pathophysiology and management of OSA. Nasal obstruction can be treated with medication, such as nasal or oral steroids, but also surgically in the presence of structural abnormalities. Surgical interventions such as septoplasty, turbinate reduction, sinus surgery and nasal valve reconstruction can help relieve nasal obstruction complaints. The effect of isolated nasal surgery on the apnea-hypopnea index (AHI) has been described in several studies. However, controversy remains about the ability of isolated nasal surgery to improve OSA. To date, three systematic reviews have described the effect of nasal surgery on the AHI. Controversially, the conclusions differ between the reviews. The most recent meta-analysis by Wu et al. (2017) described a significant reduction in the AHI after isolated nasal surgery [7]. In contrast, Li et al. and Ishii et al. reported no significant reduction in the AHI after nasal surgery [8,9]. The effect of nasal surgery on subjective outcomes such as the Epworth Sleepiness Scale (ESS) was better. The ESS improved significantly after nasal surgery in all studies. The objective of this systematic review is to give an updated overview of the literature on whether isolated nasal surgery can improve OSA subjectively (ESS) and/or objectively (polysomnography (PSG)). 

## 2. Materials and Methods

The Preferred Reporting Items for Systematic Reviews and Meta-Analyses (PRISMA) statement was followed [10]. Computerized and manual searches were performed to identify all relevant data. Studies were identified by searching PubMed, Embase.com, Cochrane Database of Systematic Reviews and CENTRAL from inception to 20 June 2022. The keywords and MeSH terms used were: “sleep apnea, obstructive,” “OSA,” “nasal surgical procedures,” “septoplasty,” “rhinoplasty,” “septorhinoplasty” and “turbinectomy.” The complete search strategies for all databases can be found in Appendix A. The computerized search yielded a total of 2486 studies, 1954 after the removal of duplicates. Two investigators independently assessed the titles and abstracts for eligibility (Figure 1). The investigators discussed any disparities regarding the inclusion of studies, and based on the inclusion criteria, a joint decision was made. The reference lists of the included studies and previous reviews were checked to ensure that additional relevant studies were not missed.

### 2.1. Inclusion/Exclusion Criteria 

Only articles with full text availability were evaluated in accordance with the inclusion and exclusion criteria. The inclusion criteria were: patients with OSA (defined as AHI ≥ 5 events/h); pre- and postoperative objective changes in AHI (obtained from level 1 or 2 polysomnography) or subjective changes in the ESS, isolated nasal surgery treatment including septorhinoplasty, rhinoplasty, nasal valve reconstruction, sinus surgery or varying methods of turbinate reduction in absence of any other level surgery. The exclusion criteria were: age < 18 years, patients without OSA or only snoring, multilevel surgery, pre- and postoperative outcomes derived from home sleep test with type 3 or 4 sleep study, review articles, case reports, conference abstracts and non-English published articles. 

### 2.2. Data Extraction 

Data were extracted to determine study methodology, results and conclusions. The published data from the included studies were reported in a worksheet by one of the authors and checked by the other authors to ensure accuracy. Study characteristics including study design and study duration were extracted. Details on patient characteristics, type of surgical intervention, outcomes (AHI/ESS), continuous positive airway pressure (CPAP) usage during the study, and data conclusions were also extracted. 

### 2.3. Data Analysis 

Descriptive analysis was performed on the pooled data using means and standard deviations when available. Statistical meta-analysis was conducted using R statistical software version 4.2.0 (R Foundation for Statistical Computing, Vienna, Austria). Study means and mean differences (postoperative—preoperative) for AHI and ESS were calculated. Accompanying 95 percent confidence intervals were calculated when standard deviations were available. The *I*^2^ test was used to calculate the heterogeneity of the studies. A random-effects model was used according to the DerSimonian and Laird method for mean differences. Study designs, patient groups and types of surgeries varied between studies, and therefore, conclusions from this analysis should be drawn with caution.

### 2.4. Quality Assessment of Included Studies

The NIH Quality Assessment Tool for Observational Cohort and Cross-Sectional Studies was used to perform a quality assessment of the included studies [11]. This tool helps to evaluate the internal validity and the potential for bias in different studies. Study designs not matching this assessment tool were separately evaluated with more design-specific quality assessment questions. The answers to the questions will be summarized.

## 3. Results 

### 3.1. Study Characteristics

A total of 1954 titles and abstracts were identified in the search. The study selection flow chart is shown in Figure 1. Twenty-seven full-text studies were assessed for eligibility. Six studies were excluded because the outcome parameters did not report the AHI or a home sleep test with a type 3 or 4 sleep study was used to diagnose OSA. A total of 21 studies met our inclusion criteria and were included in the final analysis [12,13,14,15,16,17,18,19,20,21,22,23,24,25,26,27,28,29,30,31,32]. One randomized controlled trial, 1 nonrandomized controlled trial, 15 prospective studies and 4 retrospective studies were included. Polysomnography was performed in all studies to assess OSA outcomes. Several studies divided OSA treatment outcomes into subgroups based on AHI, where mild OSA was defined as AHI between 5 and 15 events/h, moderate OSA as AHI 15–30 events/h and severe OSA as AHI > 30 events/h. Nasal surgery was performed in all studies and consisted of septoplasty, the submucous resection of the septum, septorhinoplasty, turbinate reduction performed differently, endoscopic sinus surgery and nasal valve surgery. Subjective outcomes were mostly assessed with the ESS. The study protocols are shown in Table 1. 

### 3.2. Patients Characteristics 

The majority of the included patients were male; the percentages of males in the various studies ranged from 52.96% to 100.00%. The age of included patients varied from 18 to 70 years. The details per study are shown in Table 2. 

### 3.3. Apnea–Hypopnea Index/Epworth Sleepiness Scale 

The overall AHI was not significantly different after nasal surgery in twelve studies [15,16,17,18,21,22,23,24,25,28,29,30]. In one study, the AHI only significantly decreased in the mild OSA group, defined as AHI between 5 and 15 events/h [14]. In the remaining eight studies, the overall AHI decreased significantly after nasal surgery [12,13,19,20,26,27,31,32]. Table 2 shows the exact results per study. Three studies were not included in the meta-analysis for change in AHI because no standard deviation of outcome was provided [15,20,30]. Wu et al. only provided the AHI for different subgroups; therefore, the subgroups were separately analyzed [14]. In total, 18 studies were analyzed. There was a considerable clinical heterogeneity between studies, with *I*^2^ = 98%. There was an overall decrease of 4.08 in the mean difference between preoperative and postoperative AHI. The forest plot of the random-effects model for the meta-analysis is shown in Figure A1 in the Appendix A. 

The ESS was significantly lower after nasal surgery in eighteen studies; the remaining three studies [12,13,20] did not report the ESS, as shown in Table 2. Six studies were not included in the meta-analysis for change in ESS because no standard deviation of outcome was provided [12,13,14,19,20,23]. The heterogeneity test resulted in considerable heterogeneity between studies, with *I*^2^ = 100%. There was an overall decrease of 4.01 in the mean difference between preoperative and postoperative ESS as shown in Figure A2 in Appendix A.

### 3.4. CPAP Usage

In thirteen studies, it was not reported if patients used CPAP during the study period [12,13,14,15,16,19,20,23,24,26,27,29,32]. Nakata et al. included patients using CPAP, but it was unclear if they used CPAP postoperatively [21]. The only randomized controlled trial excluded patients with CPAP therapy during the course of the study [25]. Three studies specifically evaluated the effect of nasal surgery in patients with CPAP intolerance [22,28,30]. In these studies, almost all patients tolerated CPAP postoperatively. In another study, CPAP titration was carried out in all patients, and postoperatively, there was a reduction in CPAP titration pressure, but it was not significant [18]. Tagaya et al. compared a nasal surgery group to a CPAP therapy group and according to daytime sleepiness, nasal surgery was more satisfactory for OSA patients than CPAP therapy [17]. Another study evaluated the effect of nasal surgery between CPAP users and non-CPAP users and found that CPAP users had a lower ESS after nasal surgery compared with non-CPAP users [31].

### 3.5. Quality Assessment Individual Studies 

A detailed overview of the quality assessment is shown in Table 3. The randomized trial is not shown in the table because the quality assessment questions were not applicable to this study [25]. More study-design-specific quality assessment questions were applied. This study clearly stated randomization, blinding and reason for dropout. Additionally, sufficient sample size was reported to be able to detect a difference with 80% power. Overall, this study had a good quality rating. Although Tagaya et al. and Xiao et al. performed no observational cohort studies, they are summarized in Table 3, which still gives a good summary of the quality. Overall, studies clearly stated their research question and specified the selection criteria for their study population except one study [31]. Patient selection was not clearly specified in this study. One study did not specify the type of nasal surgery [15], and five studies did not specify the time frame within which patients were included [20,23,24,27,29]. Most studies were prospective and clearly stated the in- and exclusion criteria. In one study, only 56 out of 156 eligible patients were included because there was no postoperative PSG, making this study susceptible to selection bias [12]. OSA was diagnosed prior to nasal surgery with PSG, and a postoperative PSG was performed after an acceptable time. Obstructive respiratory events were consistently scored according to the American Academy of Sleep Medicine criteria. In one study, it was not clear whether PSG was used as a preoperative measurement [32]. Two studies did not report a time frame between preoperative and postoperative PSG [17,21]. For all studies, question 8 was not applicable because all patients received nasal surgery. Some studies reported on possible confounders; only three studies adjusted statistically for their impact [12,15,19]. Therefore, it is unclear if the impact of nasal surgery is free of bias in the rest of the studies. 

## 4. Discussion

Nasal surgery can be seen as a part of the comprehensive care of OSA. The purpose of this review was to give an updated systematic review of the literature and quality assessment of the current literature on whether isolated nasal surgery can improve OSA subjectively and/or objectively. The majority of included studies suggest that nasal surgery does not significantly improve OSA in terms of AHI; however, the meta-analysis suggests that the AHI decreases slightly after nasal surgery. The significance of this reduction is questionable because of the considerable heterogeneity between studies. In terms of subjective outcomes measured with the ESS, almost all studies reported a significant improvement after nasal surgery.

Several studies suggest that nasal surgery plays an adjunctive part in the management of OSA [33]. For example, Tsai et al. stated that the multimodality treatment and holistic care of OSA should involve nasal surgery to optimize treatment outcomes [34]. However, the current literature remains controversial about the ability of isolated nasal surgery to improve respiratory disturbances in OSA. This review found similar results to the most recent published meta-analysis. Wu et al. concluded that there was a significant improvement in the AHI and ESS after isolated nasal surgery [7]. Meanwhile, two previous systematic reviews and a meta-analysis concluded that the AHI did not significantly decrease after nasal surgery [8,9]. These studies did find similar results for the improvement in ESS. 

When interpreting these conclusions, it needs to be considered that there was a considerable clinical heterogeneity between studies. This review therefore tried to focus on the quality of the included studies and possible explanations for controversies and discrepancies in the current literature. Variations in study designs and the lack of large randomized trials contribute to the heterogeneity. Only one RCT met the inclusion criteria [25]. The remaining studies were either prospective or retrospective cohort studies, being more susceptible to selection bias. Notable is that this RCT and the relatively larger studies of higher quality more often found that nasal surgery does not improve OSA in terms of AHI. Patient selection also was different for many studies; several studies included all severities of OSA, but other studies only included patients with severe OSA and CPAP failure. The causes of nasal obstruction also differed between studies; some studies included patients with allergic rhinitis, and others only included patients with structural abnormalities. In addition, the types of nasal surgeries were inconsistent, and this also contributed to the considerable heterogeneity. On the other hand, by only including studies using level 1 or 2 polysomnography to determine OSA severity, heterogeneity was reduced. 

This review does have several limitations. Because of this considerable heterogeneity, no proper significant meta-analysis with subgroup analysis was performed. Although a previous meta-analysis did succeed in demonstrating significant results, no relevant subgroup analysis was performed. In order to prevent misleading conclusions, this review focused on descriptive analysis and not meta-analysis. The clinical relevance of the significant decrease of AHI in the eight studies included in this review is also questionable. The decrease was often small and did not often result in treatment success as defined according to Sher [35], with a reduction of 50% in AHI and an AHI less than 20. Additionally, the overall postoperative decrease in AHI of 4.08 in the mean difference will not result in treatment success. The different times to follow-up will also have contributed to the inter-study differences in changes in AHI. Most studies repeated the polysomnography after three months postoperatively, but Kim et al., for example, found a more significant reduction in the AHI after six months [26]. The longest mean period of follow-up was 12 months. 

In almost all studies, subjective outcomes improved significantly after nasal surgery. The significant improvement in ESS is also questionable because of the considerable heterogeneity between studies. This could possibly be explained because different patients were selected for nasal surgery, and CPAP was used simultaneously in some studies. Bican et al. and Elwany et al. concluded that patients tolerated CPAP therapy better after nasal surgery [28,31]. They found a significant decrease in CPAP titration pressure after nasal surgery. Better tolerance resulting in better sleep quality could help explain the improvement in ESS in some studies rather than the nasal surgery alone. On the contrary, Tagaya et al. found that the improvement in ESS was more significant for nasal surgery compared with CPAP therapy in OSA patients [17]. 

There is a remarkable discrepancy between the objective and subjective results of isolated nasal surgery. Patients often feel better, but this is usually not reflected in a substantial improvement in the AHI, suggesting that nasal surgery could be an adjunct to OSA treatment based on subjective outcomes. A meta-analysis also described that isolated nasal surgery reduces therapeutic CPAP pressure in patients with OSA and nasal obstruction [36]. Therefore, nasal surgery could contribute to better CPAP compliance in patients with OSA.

## 5. Conclusions

Based on the present analysis of objective outcomes, isolated nasal surgery did not improve the AHI significantly in the majority of the studies. The meta-analysis suggests a slight decrease in AHI after nasal surgery, but this reduction is not clinically relevant in terms of treatment success. Isolated nasal surgery should therefore not be recommended as a first-line treatment for OSA. Because of high heterogeneity, these results should be interpreted with caution. Isolated nasal surgery can possibly improve OSA subjectively. Perhaps only OSA patients with complaints of nasal obstruction or OSA patients experiencing difficulty with CPAP compliance would benefit from nasal surgery. More consistent randomized trials should be performed to evaluate the effect of isolated nasal surgery on OSA in specific subgroups.

## Figures and Tables

**Figure 1 brainsci-12-01446-f001:**
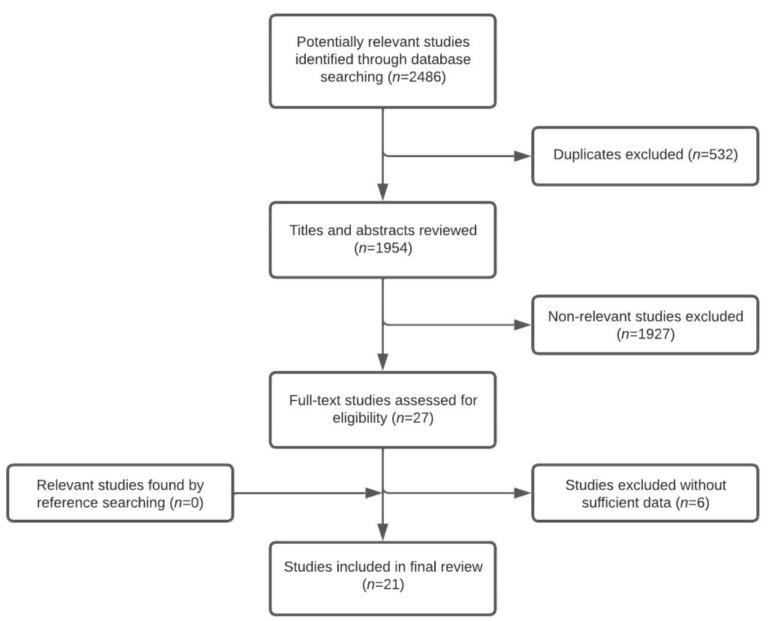
Flow chart of study selection.

**Table 1 brainsci-12-01446-t001:** Study characteristics.

Author and Year	Study DeSign (*n* = Sample Size)	Intervention	Outcomes	Study Time *	Conclusion
Li et al., 2022 [22]	Prospective (*n* = 30)	Septoplasty, the lateralization of the inferior turbinate, middle turbinoplasty, uncinectomy, expansion of the maxillary ostia, bilateral total ethmoidectomy	AHI, CT images ESS, SOS, VAS	3 months	Nasal surgery did not significantly reduce the AHI but decreased the total resistance of the upper airway and increased the nasal airflow volume and ESS in patients with nasal obstruction and OSA.
Elwany et al., 2022 [28]	Prospective (*n* = 49)	Septoplasty and radiofrequency turbinate reduction, powered turbinoplasty, radiofrequency turbinate reduction	AHI, CPAP pressure ESS, NOSE score	3–6 months	Nasal surgery resulted in a significant decrease in mean ESS, NOSE score and CPAP titration pressure, but no significant decrease in AHI was found.
Wu et al., 2022 [14]	Prospective (*n* = 100)	Septorhinoplasty, the internal displacement fixation of middle turbinate fracture, the external displacement fixation of inferior turbinate fracture, and the symmetrical opening of the bilateral sinuses in the middle nasal canal	AHI, nasal resistance ESS, SOS, SBPS, VAS, SNOT-20 scores	6 months	Nasal surgery can effectively improve subjective outcomes and reduce nasal resistance. Only in patients with mild OSA did the AHI decrease significantly after surgery.
Kim et al., 2021 [26]	Prospective (*n* = 35)	Septoplasty and inferior turbinate reduction	AHI ESS, VAS, NOSE	6 months	Isolated nasal surgery improves AHI and subjective outcomes in patients with OSA and severe nasal obstruction.
Bosco et al., 2020 [30]	Prospective (*n* = 34)	Septoplasty and turbinoplasty	AHI, NOHL scale (DISE) ESS	3 months	The upper airway obstruction pattern changed significantly after nasal surgery. The ESS was significantly lower, and the AHI decreased after surgery, but the difference was not significant.
Abd El-Aziz et al., 2018 [32]	Prospective (*n* = 60)	Septal correction combined with partial turbinectomy or septoplasty alone	AHI ESS	3 months	Intranasal surgery has a significant effect on AHI and subjective quality of sleep in selected OSA patients with both septal deviation and hypertrophy of the inferior turbinates.
Tagaya et al., 2017 [17]	Retrospective case-control (*n* = 40/40)	Inferior turbinectomy or the submucous resection of the nasal septum (or both) or endoscopic sinus surgery	AHI ESS	-	Nasal surgery did not decrease the AHI, but it significantly improved the quality of sleep. The improvement in the ESS was more significant in the surgery group compared with the CPAP group.
Xiao et al., 2016 [13]	Nonrandomized controlled (*n* = 30/30)	Septoplasty, the medial displacementand fixation of the middle turbinate, sinus surgery, the lateral displacement and fixation of the inferior turbinate	AHI PSQI, SCL-90, VAS, GSI	3 months	Nasal surgery significantly decreases the severity of OSA in patients with nasal obstruction.
Hisamatsu et al., 2015 [27]	Prospective (*n* = 45)	Compound nasal surgery: conventional septoplasty and submucosal inferior turbinectomy with posterior nasal neurectomy	AHI ESS, QOL	3 months	Compound nasal surgery significantly improves subjective and objective OSA events in selected patients.
Shuaib et al., 2015 [19]	Retrospective (*n* = 26)	Functional septoplasty, subtle aesthetic refinement	AHI ESS, NOSE	3 months	Functional rhinoplasty may have the potential to significantly improve OSA severity in selected patients with nasal obstruction.
Yalamanchali et al., 2014 [12]	Retrospective (*n* = 56)	Septoplasty with bilateral submucosal inferior turbinate reduction and concurrent endoscopic sinus surgery, nasal valve repair and nasal polypectomy	AHI	12 months	The median AHI decreased significantly after nasal surgery and sinus surgery in patients with moderate and severe OSA.
Victores et al., 2012 [15]	Retrospective (*n* = 24)	Nasal surgery	AHI ESS	3 months	The pattern of upper airway obstruction and AHI did not significantly change after nasal surgery.
Sufioğlu et al., 2012 [18]	Prospective (*n* = 31)	Septoplasty, septorhinoplasty, septoplasty and RFA inferior turbinate, endoscopic sinus surgery and septoplasty and RFA inferior turbinate, RFA inferior turbinates bilaterally	AHI, CPAP-titration ESS, VAS, OSAS complaint questionnaire	3 months	After nasal surgery alone, subjective complaints improved, but the AHI did not significantly improve.
Choi et al., 2011 [29]	Prospective (*n* = 22)	Endoscopic sinus surgery, septal surgery, turbinate surgery	AHI ESS	3 months	Nasal surgery can partially improve ESS and snoring, but it had no effect on AHI in patients with OSA and nasal obstruction.
Bican et al., 2010 [31]	Prospective (*n* = 20)	Septoplasty, nasal valve reconstruction, concha cauterization	AHI, CPAP level ESS	117 days (range, 80–189 days)	The AHI and ESS decreased significantly after nasal surgery in patients with OSA, and tolerating the CPAP device was easier.
Li et al., 2009 [24]	Prospective, parallel study (*n* = 66)	Septoplasty and the volume reduction of inferior turbinates	AHI ESS, SOS	3 months	Nasal surgery relieves snoring and improves the ESS, but it had variable effects on objective outcomes.
Koutsourelakis et al., 2008 [25]	Randomized controlled (*n* = 27/22)	The resection of deviated septum, or with resection bilateral inferior turbinates	AHI, nasal resistance ESS	3–4 months	Nasal surgery rarely treats OSA effectively. Nasal breathing epochs can predict surgery outcome.
Nakata et al., 2008 [21]	Prospective (*n* = 49)	The submucous resection of the nasal septum or inferior turbinectomy (or both) or endoscopic sinus surgery	AHI ESS	-	Nasal surgery can lower nasal resistance and ESS but did not improve the AHI.
Li et al., 2008 [23]	Prospective (*n* = 51)	The resection of the bowed septum and the excision of the lateral part of the inferior turbinate (septomeatoplasty)	AHI ESS, SOS, SF-36	3 months	Nasal surgery in patients with nasal obstruction and OSA improves disease-specific quality of life, snoring and ESS.
Verse et al., 2002 [16]	Prospective (*n* = 26)	Septorhinoplasty, septoplasty, septoplasty with bilateral paranasal sinus surgery, nasal tip/nasal valve surgery	AHI, nasal resistance ESS	12 months (range, 3–50 months)	Nasal surgery has limited efficacy in the treatment of patients with OSA. Nasal surgery can significantly improve sleep quality and daytime sleepiness.
Sériès et al., 1993 [20]	Prospective (*n* = 14)	Septoplasty, turbinectomy, and polypectomy	AHI, nasal resistance	2–3 months	Nasal surgery significantly improved the AHI in patients with OSA.

AHI = apnea-hypopnea index, CPAP = continuous positive airway pressure, DISE = drug-induced sleep endoscopy, ESS = Epworth Sleepiness Scale, GSI = Global Symptom Index, NOHL scale = Nasopharynx cavity and walls, Oropharynx, Hypopharynx, Larynx scale, NOSE = Nasal Obstruction Symptom Evaluation, OSA = obstructive sleep apnea, PSQI = Pittsburgh Sleep Quality Index, RFA = radiofrequency ablation, SBPS = Spouse/Bed Partner Survey, SCL-90 = Symptom Check List 90, SF-36 = 36-item Short-Form Health Survey, SNOT-20 = Sino-Nasal Outcome Test, SOS = snores outcomes survey, VAS = visual analogue scale. * time between operation and follow-up PSG.

**Table 2 brainsci-12-01446-t002:** Patients’ characteristics.

Author and Year	Sample Size (*n*)	Sex (Male %)	Age, Year	Apnea–Hypopnea Index			Epworth Sleepiness Scale		
				AHI Preoperative	AHI Postoperative	Significance	ESS Preoperative	ESS Postoperative	Significance
Li et al., 2022 [22]	30	83.30	32.4 (13.2) ^a^/41.8 (8.2) ^b^/45.1 (9.4) ^c^	24.5 (13.7)	22.7 (14.6)	NS (*p* = 0.492)	12.7 (1.2)	8.6 (2.9)	*p* < 0.000
Elwany et al., 2022 [28]	49	75.50	52.2 (9.54)	48.9 (2.2)	44.1 (1.4)	NS	10.45 (1.67)	4.98 (0.88)	*p* < 0.001
Wu et al., 2022 [14]	100	92.00	35.8 (11.7)	12.1 (3.7) ^d^ 24.5 (4.5) ^e^ 51.1 (10.3) ^f^	7.1 (2.6) ^d^ 22.6 (4.5) ^e^ 50.6 (14.5) ^f^	*p* < 0.05 ^d^ NS ^e,f^	4.4 (4.9) ^d^ 4.8 (5.2) ^e^ 3.9 (3.9) ^f^	-	*p* < 0.01 ^d,e,f^
Kim et al., 2021 [26]	35	97.14	42.8 (13.6)	28.5 (22.3)	18.5 (19.8)	*p* < 0.001	7.9 (4.9)	5.3 (3.8)	*p* < 0.001
Bosco et al., 2020 [30]	34	67.65	42.8 (14)	26.7 (22.4)	19	NS (*p* > 0.05)	8.4 (5)	6.5 (5)	*p* < 0.05
Abd El-Aziz et al., 2018 [32]	30 ^g^/30 ^h^	56.60 ^g^/70.00 ^h^	35.5 (9.9) ^g^/39.0 (7.8) ^h^	14.21 (4.56)	12.29 (4.63)	*p* = 0.001	12.02 (2.46)	11.15 (2.64)	*p* < 0.001
Tagaya et al., 2017 [17]	40	100.00	48.1 (11.3)	52.6 (18.9)	49.5 (17.8)	NS (*p* = 0.11)	11.0 (4.0)	5.1 (2.3)	*p* < 0.001
Xiao et al., 2016 [13]	30	100.00	45.5 (11.37)	49.67 (19.49)	43.07 (21.86)	*p* < 0.01	-	-	-
Hisamatsu et al., 2015 [27]	43 *	93.02	48.25 (11.39)	51.06 (17.71)	38.60 (17.75)	*p* < 0.05	14.76 (2.39) **	8.06 (3.33) **	*p* < 0.0001
Shuaib et al., 2015 [19]	26	65.38	42.7 (13.6)	24.7 (18.8)	16 (16.1)	*p* = 0.013	11.5	7.5	*p* = 0.003
Yalamanchali et al., 2014 [12]	56	85.71	43.6 (11.3)	33.5 (22)	29.4 (20.8)	*p* = 0.009	-	-	-
Victores et al., 2012 [15]	24	79.17	44.8 (13.9)	27.3 (18.1)	24.4	NS (*p* > 0.05)	12.3 (6.2)	6.6 (4.2)	*p* < 0.05
Sufioğlu et al., 2012 [18]	28	83.90 ***	53 (9.6)	32.5 (22.6)	32.4 (24.6)	NS (*p* = 0.69)	9.3 (5.1)	5.9 (3.9)	*p* < 0.001
Choi et al., 2011 [29]	22	100.00	41.3 (10.9)	28.9 (20.4)	26.1 (21.9)	NS (*p* = 0.445)	8.8 (3.3)	6.3 (3.3)	*p* = 0.001
Bican et al., 2010 [31]	20	100.00	47.5	43.1 (27.1)	24.6 (22.2)	*p* < 0.05	17.1 (2.7)	11.1 (2.8)	*p* < 0.01
Li et al., 2009 [24]	44	95.45	38.3 (9.9)	36.4 (29.1)	37.5 (31.6)	NS	10.6 (3.9)	7.6 (4.5)	*p* = 0.02
Koutsourelakis et al., 2008 [25]	27	52.96	39 (7.5)	31.5 (16.7)	31.5 (18.2)	NS	13.4 (2.9)	11.7 (3.4)	*p* < 0.01
Nakata et al., 2008 [21]	49	100.00	46.1 (12.3)	44.6 (22.5)	42.5 (22)	NS	10.6 (4.1)	4.5 (2.6)	*p* < 0.001
Li et al., 2008 [23]	51	98.04	39 (10)	37.4 (28.9)	38.1 (32.7)	NS	10	8	*p* < 0.001
Verse et al., 2002 [16]	26	96.15	52.5 (8.4)	31.57 (25.6)	28.93 (24.73)	NS	11.87 (4.7)	7.73 (4.96)	*p* < 0.004
Sériès et al., 1993 [20]	14	85.71	30–58 ****	17.0	6.5	*p* < 0.025	-	-	-

Data are presented as mean and (standard deviation), AHI = *apnea-hypopnea index*, ESS = *Epworth Sleepiness Scale*, NS = *not significant.*
^a^ Mild OSA group ^b^ moderate OSA group ^c^ severe OSA group; ^d^ Mild OSA group (*n* = 26) ^e^ moderate OSA group (*n* = 24) ^f^ severe OSA group (*n* = 27), the ESS in these groups is shown as mean difference before and after surgery; ^g^ Group with septoplasty with inferior turbinate reduction ^h^ group with only septoplasty; * only patients with severe OSA; ** results ESS from *n* = 33; *** 31 patients analysis with 28 patients; **** range of age.

**Table 3 brainsci-12-01446-t003:** Quality assessment individual studies.

	Li et al., 2022 [22]	Elwany et al., [28]	Wu et al., [14]	Kim et al., [26]	Bosco et al., [30]	Abd El-Aziz et al., [32]	Tagaya et al., [17]	Xiao et al., [13]	Hisamatsu et al., [27]	Shuaib et al., [19]	Yalamanchali et al., [12]	Victores et al., [15]	Sufioğlu et al., [18]	Choi et al., [29]	Bican et al., [31]	Li et al., 2009 [24]	Nakata et al., [21]	Li et al., 2008 [23]	Verse et al., [16]	Sériès et al., [20]
1. Was the research question or objective in this paper clearly stated?	Yes	Yes	Yes	Yes	Yes	Yes	Yes	Yes	Yes	Yes	Yes	Yes	Yes	Yes	Yes	Yes	Yes	Yes	Yes	Yes
2. Was the study population clearly specified and defined?	Yes	Yes	Yes	Yes	Yes	Yes	Yes	Yes	Yes ^d^	Yes	Yes	Yes ^g^	Yes	Yes ^d^	No ^h^	Yes ^d^	Yes	Yes ^d^	Yes	Yes ^d^
3. Was the participation rate of eligible persons at least 50%?	Yes	Yes	Yes	Yes	Yes	Yes	Yes	Yes	Yes	Yes	No ^f^	Yes	Yes	Yes	Yes	Yes	Yes	Yes	Yes	Yes
4. Were all the subjects selected or recruited from the same or similar populations (including the same time period)? Were inclusion and exclusion criteria for being in the study prespecified and applied uniformly to all participants?	Yes	Yes	Yes	Yes	Yes	Yes	Yes	Yes	Yes	Yes	Yes	Yes	Yes	Yes	Yes	Yes	Yes	Yes	Yes	Yes
5. Was a sample size justification, power description, or variance and effect estimates provided?	No	No	No	No	No	No	No	No	No	No	No	No	No	No	No	No	No	No	No	No
6. For the analyses in this paper, were the exposure(s) of interest measured prior to the outcome(s) being measured?	Yes	Yes	Yes	Yes	Yes	Yes	Yes	Yes	Yes	Yes	Yes	Yes	Yes	Yes	Yes	Yes	Yes	Yes	Yes	Yes
7. Was the timeframe sufficient so that one could reasonably expect to see an association between exposure and outcome if it existed?	Yes	Yes	Yes	Yes	Yes	Yes	NR ^c^	Yes	Yes	Yes	Yes	Yes	Yes	Yes	Yes	Yes	NR ^c^	Yes	Yes	Yes
8. For exposures that can vary in amount or level, did the study examine different levels of the exposure as related to the outcome (e.g., categories of exposure, or exposure measured as continuous variable)?	NA	NA	NA	NA	NA	NA	NA	NA	NA	NA	NA	NA	NA	NA	NA	NA	NA	NA	NA	NA
9. Were the exposure measures (independent variables) clearly defined, valid, reliable, and implemented consistently across all study participants?	Yes	Yes	Yes	Yes	Yes	Yes	Yes	Yes	Yes	Yes	Yes	Yes	Yes	Yes	Yes	Yes	Yes	Yes	Yes	Yes
10. Was the exposure(s) assessed more than once over time?	No	No	No	No	No	No	No	No	No	No	No	No	No	No	No	No	No	No	No	No
11. Were the outcome measures (dependent variables) clearly defined, valid, reliable, and implemented consistently across all study participants?	Yes	Yes	Yes	Yes	Yes	No ^b^	Yes	Yes	Yes	Yes	Yes	Yes	Yes	Yes	Yes	Yes	Yes	Yes	Yes	Yes
12. Were the outcome assessors blinded to the exposure status of participants?	NA	NA	NA	NA	Yes ^a^	NA	NA	NA	NA	NA	NA	NA	NA	NA	NA	NA	NA	NA	NA	NA
13. Was loss to follow-up after baseline 20% or less?	Yes	Yes	Yes	Yes	Yes	Yes	NA	Yes	Yes	NA	NA	NA	Yes	Yes	Yes	Yes	Yes	Yes	Yes	Yes
14. Were key potential confounding variables measured and adjusted statistically for their impact on the relationship between exposure(s) and outcome(s)?	No	No	No	No	No	No	No	No	No	Yes ^e^	Yes ^e^	Yes ^e^	No	No	No	No	No	No	No	No

NA not applicable NR not reported; ^a^ The video recordings were analyzed by someone who was blinded to the data. ^b^ Unclear if postoperative PSG outcomes are compared with polysomnography outcomes or Embletta PDS device outcomes. ^c^ Unclear time frame between preoperative and postoperative outcomes. ^d^ The study population clearly specified but no time frame was given. ^e^ Analysis adjusted for BMI. ^f^ From the 156 eligible patients, only 56 were included. ^g^ Type of nasal surgery not specified. ^h^ Unclear if only patients with OSA or patients with complaints of hypersomnia and snoring are included and no time frame was given. The one RCT was not included in this quality assessment (Koutsourelakis et al.).

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
