# Peer review of "The Role of Isolated Nasal Surgery in Obstructive Sleep Apnea Therapy—A Systematic Review"

_brainsci, 2022, doi:10.3390/brainsci12111446_

Round 1

Reviewer 1 Report

The authors have performed a systematic review on the effect of isolated nasal surgery on the AHI and the ESS in OSA.

Major comments:

Authors state in their discussion that studies were too heterogeneous to do a meta-analysis. How was this determined? Is this due to the different types of surgery in each study and potentially in each patient? One could still argue that a meta-analysis with presentation of a forest plot with post vs. pre procedure AHI and ESS could help understand results, notwithstanding caveats that do need to be discussed. As such, the previous systematic review (ref 7) was able to do a meta-analysis. Here, authors excluded studies using HSAT rather than laboratory PSG for AHI assessment, which reduced heterogeneity in that regard. The current list of studies adds relatively little to the literature, even if it includes more recent studies, because it does not provide a summary measure, and it does not attempt to understand how the heterogeneity might affect results.

A limitation regarding the ESS assessment months after the surgery is that it is not clear if patients were using CPAP or not, and if adherence improved after surgery. It should be included in the data tables, if patients were on CPAP or not, or not reported. This has major implications for the ESS outcome. Lower pressures with possibly lower adverse effects (e.g. leak dry mouth, etc), better tolerance, and better sleep quality could help explain the improved ESS and should not be discounted. The discussion relates to this in the last paragraph, but authors do not attempt to evaluate this in their systematic review. More information could perhaps be extracted as part of their review to help support their conclusion that “Perhaps only OSA patients with complaints of nasal obstruction or OSA patients experiencing difficulty with CPAP compliance would benefit from nasal surgery”. Subgroup analyses could potentially be done which would really increase the value of the paper.

Minor:

 P 3, line 131: Measurement instruments were good for all studies – please clarify measurement of what – PSG equipment?

Suggest to re-phrase this sentence for clarity – line 221 on first page of discussion: It is however, important to keep into consideration that patients undergoing nasal surgery are a lot, more likely to be scheduled for nasal surgery than those without nasal complaints.

Line 203: “Therefore, there is a clear selection bias when reviewing subjective outcomes.” It is not surprising that patients with complaints are selected for surgery. One would not operate on a patient without nasal obstruction. I am not sure this constitutes a selection bias in relation to ascertainment of outcome. Rather, it defines the population studied. There might be bias if the more sleepy people are targeted for surgery, but even then it is a question of defining your target population. There is difficulty in interpreting results as there is lack of a control group in most studies so a placebo effect if definitely not excluded.

Reviewer 2 Report

1. Among the studies reviewed by Wu et al. (2017, systematic review), papers of Park et al. (2014) and Nakata et al. (2005) are available in PubMed and seem to satisfy the inclusion criteria of your study. Please provide reasons why they were not included in the present systematic review.

2. In the systematic review study published in 2017 (Wu et al.), a meta-analysis was performed with 18 papers. Please provide specific rationales for why a meta-analysis could not be conducted. 

3. The discussion section is short. In the discussion, the studies were described as not being meta-analyzed because they were too heterogeneous. Please suggest what the rationale for the heterogeneity (data review content, forest plot, funnel plot, etc.) is.

Reviewer 3 Report

The manuscript entitled "The Role of Isolated Nasal Surgery in Obstructive Sleep Apnea Therapy – a systematic review" aimed to compare AHI and ESS for isolated nasal surgery treated OSA among the overview of the literatures.

The authors had stated the goals and reported the inclusion/exclusion criteria. They also concluded that isolated nasal surgery cannot reduce AHI significantly, but it can decrease ESS.

The below was my concern:

1. The enrolled references were important materials for system review. Citations with parentheses may be confused to other symbols; for example, reference (30) and sample (30). Please improve.

2. Please add some description for NIH quality assessment tool in 2.3

3. A total of 21 studies were met the inclusion criteria, were they [12-32] rather than [11-31] (line 99)? Based on the error, the report in 3.3 for apnea-hypopnea index/ ESS should be checked for references with 21-32 (line 116) and 12, 15, 33 (line 120).

4.Since the AHI and ESS were indictors for assessment, why Series et al. 1993 was without ESS in Table 2?

5. There were series errors of references presented in Table 3. For example, We et al. was [20] rather than [29]. Please updated. Besides, one reference was missing.

6. There were of 9 reports was significant difference for AHI in Table 2. It took 42.9 % (9/21) improvement. How the authors concluded the nasal surgery does not improve OSA in terms of AHI (line 174)? Was the measurement time different? (3 or 12 months?) Moreover, some patients with mild OSA may care the daytime sleepiness, how to judge isolated nasal surgery cannot be recommended as first line treatment of OSA?

Round 2

Reviewer 1 Report

No further comments.

Reviewer 2 Report

It seems that the comments have been reflected in the manuscript, and it has been well-revised.

I have a few additional comments to improve the quality of the paper.

1. Please, describe in detail in ‘2. Materials and methods – 2.1. inclusion/exclusion criteria. example) Which level of full-night polysomnography was included? or Other devices such as HSAT, watch-PAT, and Reggie device were excluded as your response (We also tried to reduce heterogeneity by only having studied using PSG and excluded studies using HSAT, watch-PAT, Reggie device, or other devices used to determine the AHI.) 

2. Verification of overall English expression is required.

Reviewer 3 Report

Please change citations # ref with parentheses “( )” to square brackets “[ ]” for the manuscript and Tables. For example: “Nasal obstruction is common in patients with obstructive sleep apnea (OSA) and is a known risk factor (1)”, (1) should be as [1].  The sample sizes please use parentheses “(n=…)” to replace square brackets “[n=…]” in Table 1. Other changes made were fine.
